# Resemblance of nutrient intakes in three generations of parent-offspring pairs: Tehran lipid and Glucose Study

Parvin Mirmiran[1], Asiyeh Sadat Zahedi[2], Glareh Koochakpour[3], Firoozeh Hosseini-Esfahani[1]*, Mahdi Akbarzadeh[2], Maryam S Daneshpour[2], Fereidoun Azizi[4]

1 Nutrition and Endocrine Research Center, Research Institute for Endocrine Sciences, Shahid Beheshti University of Medical Sciences, Tehran, Iran, 2 Cellular and Molecular Endocrine Research Center, Research Institute for Endocrine Sciences, Shahid Beheshti University of Medical Sciences, Tehran, Iran, 3 Maragheh University of Medical Sciences, Maragheh, Iran, 4 Endocrine Research Center, Research Institute for Endocrine Sciences, Shahid Beheshti University of Medical Sciences, Tehran, Iran

* f.hosseini@sbmu.ac.ir

**Data Availability Statement:** The datasets generated and analyzed during the current study are not publicly available because data contains

## Abstract

The degree of maintaining nutrient intake patterns, conformed in the family, for offspring into adulthood is unknown. The aim of this study was to investigate the correlation between nutrient intakes in three younger-middle-older generations of Tehranian adults by sex. Of individuals who participated in 2012–15 phase of the Tehran Lipid and Glucose Study, 1286 families (4685 subjects), who had at least two members of the family with complete data in two or three generations were entered in this cross-sectional study. The energy and nutrient intakes of parents and their young or adult offspring or grandparents-grandson/granddaughter dyads were compared. The differences were estimated using pairwise t-test and partial correlation. Data of parents with their offspring were paired based on living arrangement. There were 857 fathers (mean age: 55.4±11.1) and 1394 mothers (mean age: 50.1±11.4). The mean age of grandfathers and grandmothers were 69.4±7.9 and 63.7±8.5 respectively. The significant correlation in fathers-sons and father-daughter (living with their parents) pairs were observed for 9 and 7 nutrients, respectively. Correlations for most nutrients were significant for mother-daughter or sons (living with their parents) dyads. The mean percentage of energy from total fat and trans-fatty acids of sons or daughters (living with their parents) were higher than their parents. For most nutrients, there were no significant adjusted correlations between parents-adult offspring (living independent of their parents) dyads. Also few nutrient intakes of grandparents-grandson or granddaughter dyads were correlated. The nutrient intakes of adult offspring were not associated with their parents; this correlation for younger and older generations disappeared. There were weak to moderate correlation between nutrient intakes of parent-offspring dyads that lived with their parents. The resemblance was higher for mother-offspring than father-offspring. Overall, total fat and trans-fatty acid intakes of young offspring were higher than their parents.

sensitive individual information and data are owned by Research Institute for Endocrine Sciences, Shahid Beheshti University of Medical Sciences. Tehran, Iran Data are available from the ethics committee of the Research Institute for Endocrine Sciences, Shahid Beheshti University of Medical Sciences, Tehran, Iran, whenever data request has been sent. No.24, Arabi Street, Yemen Avenue, Chamran Highway Fax: +98 (21)22402463 Postal code: 1985717413 Email: info@endocrine.ac.ir.

**Funding:** This study was supported by Shahid Beheshti University of Medical Sciences, Tehran, Iran (Grant no. 15361). The funders had no role in study design, data collection and analysis, decision to publish, or preparation of the manuscript.

**Competing interests:** The authors have declared that no competing interests exist.

# Introduction

Parental dietary intakes influence the nutrient intakes of children regardless of child's age and sex through gene and home environment [1]. The dietary intake of children and their parents were associated in previous studies; it is possibly due to sharing meals with each other; parents can serve as a role model in shaping diet-related behaviors and affect children's preferences and attitudes towards diet [2,3]. The emotional connection between family members intensifies children's imitation of their parents [4]. The type of food purchased, method of food preparation, frequency of family meals, availability level of food groups in the household and deciding where the family goes out to eat, are all determined by the parents [5]. Children's intake of snacks, sweets, fruit, vegetables, and energy were partially correlated with the mother's intake of these foods and energy [6]. The positive correlation between the diets of mothers-child was stronger than for fathers-child [7,8]. Some healthy food groups such as fruit and vegetable had stronger correlation between mothers and children than unhealthy foods [9,10]. However these associations were different across countries with regard to dietary assessment methods and parent-child pairs [7].

The extent of the association of parental and offspring dietary intake can change when children enter adolescence, as they are less likely to participate in family dinner at home and the parent- child intake resemblance decreases [11]. This association is also weak in industrialized countries where the number of shared meals is decreasing [12] but, more research is needed in developing countries that are under nutrition transition. These finding raise this question of whether these dietary intakes conformed in the family for children or adolescents tend to track into adulthood when they marry or form a separate family. It is also unknown if such patterns persist or maintained through multiple generations. The answer to this question can predict the success rate of family-based interventions in changing dietary behaviors. Most studies have investigated the resemblance of dietary intakes between parents and children or adolescents and few studies have examined the association of dietary intakes between parents and their adult offspring [13–15] or between grandparents and their grandson/daughter [13–15]. Moreover there is little evidence on the similarity of dietary intakes among offspring and their parents in communities by living arrangements or marital status. In addition, the strength of each parent's influence on their sons or daughters was not comprehensively discussed in the literature. With that in mind, the aim of this study is to investigate the correlation between nutrient intakes in three younger-middle-older generations of Tehranian adults by sex. Findings can help elucidate intergenerational influences on dietary patterns or inform nutrition interventions targeting multigenerational extended families.

# Methods

## Study population

Participants for this study were enlisted from the Tehran lipid and glucose study (TLGS), a large-grade population and family-based cohort study implemented to resolve risk factors for non-communicable diseases in a representative sample of residents of district 13 Tehran, the capital of Iran. At first survey of the study (1999–2001), 15005 individuals aged $\geq 3$ years were selected using multistage stratified cluster random sampling and follow-up questioning was conducted in five consecutive phases: Phase 2 (2002–2005), Phase 3 (2005–2008), Phase 4 (2008–2011), Phase 5 (2012–2015) and Phase 6 (2015–2018) [16–18].

Of 12362 individuals who participated in Phase 5, a total of 7721 subjects (3590 men) completed the dietary assessment; these subjects were entered as population in this cross-sectional study.

Among them, 1286 families (4685 subjects), who had at least two members of the family with complete data were entered as the population in the current cross-sectional study. These two members include parental (father or mother) and their female or male- children or adult offspring in two generations. In addition, data of parents with their young or adult offspring were paired based on living status. Also, data of grandparents and their grandson or daughter were coupled.

The genetic data management system (Progeny Clinical Version 7) from Progeny Software (Progeny Software LLC, Delray Beach, FL) was used to stalk, manipulate, and error-checked family data pedigree details. A code was assigned to each family relationship; living together or independently with their parents was included in each person's particular code.

Ethical approval for this study was attained from the ethics committee of the Research Institute for Endocrine Sciences, Shahid Beheshti University of Medical Sciences, Tehran, Iran. All adult participants provided written informed consent before participating in this study and written consent was obtained from the parents of children and adolescents.

## Measurements

Skilled interviewers completed demographic data using the pre-tested questionnaire and face to face private interviews. Living arrangements of offspring were determined based on their marital status, as living together with their parents (not married) and living independently with their parents (married/cohabiting couple with or without children). The marriage and living independent of their parents were highly correlated in Iran [19]. Data of parents with their young or adult offspring were paired based on living status. Also data of grandparents and t heir grand-son or–daughter were coupled.

## Dietary assessment

Dietary data were gathered through face to face interviews using a valid and reliable 147-items semi-quantitative food frequency questionnaire (FFQ) [20,21]. Participants reported the usual frequency of consumption of individual food items using the standard serving sizes on a daily, weekly or monthly basis during the last year. The usual food intakes were then changed to daily intake (grams/day) and were converted as energy-adjusted terms (serving per 1000 kcal/day). Because the Iranian food composition table (FCT) is incomplete (limited to only raw materials and a few nutrients), the United States Department of Agriculture (USDA) FCT was used to analyze food composition [22]. The Iranian FCT was used as a substitute for Iranian food items, like kashk, which are not included in the USDA FCT. The Iranian FCT was used to calculate trans-fat content of foods [23]. The difference and correlation of total energy (kcal/day) and some nutrients were considered across three generations. To better compare usual nutrient intakes of children and their parents in two different age groups, nutrients were adjusted for energy intake (percentage of energy or per/1000 kcal of energy intake); e.g. including carbohydrate, starch and non-starch carbohydrate, protein, vegetable and animal protein, total fat, saturated fatty acid (SFA), mono-unsaturated fatty acid (MUFA), poly-unsaturated fatty acid (PUFA), trans-fatty acids (as percentage of energy), fiber (gr/1000 kcal/day), cholesterol (mg/day), sodium (mg/day), calcium, vitamin C, iron, zinc, and magnesium (as mg/1000 kcal/day). The recommended intakes of sodium and cholesterol are similar for all age groups. The selection of nutrients was based on dietary guidelines; these nutrients were more discussed in dietary guidelines.

## Anthropometric measurements

Weight was measured using digital scales (Seca 707) to the nearest 100 g, while the participants wear slightly clothed and without shoes. Height was measured to the nearest 0.5 cm using a

tape measure, in standing position with shoulders in normal alignment and without shoes. Waist circumference (WC) was measured to the nearest 0.1 cm using a non-flexible tape meter over light clothing, at the end of normal expiration and at the level of the umbilicus without any pressure to body surface.

### Physical activity

Physical activity was measured using the Persian-translated modifiable activity questionnaire with high reliability and relative validity. Data on the time and frequency of light, moderate, high, and very high severity activities were gathered based on the list of usual activities of daily life during the past year. Physical activity level was reported based on the metabolic equivalent/hour/week (MET/h/week) [24].

### Statistical analyses

Statistical analyses were performed using the Statistical Package for Social Sciences (version 21.0; SPSS). A two tailed P value <0.05 was used to determine statistical significance. The mean±SD and proportion of characteristics and dietary nutrient intakes of participants were measured. Paired t-test was used to determine the difference of energy and energy adjusted nutrient intakes of parents and their young or adult offspring stratified by living together or independently with their parents. Results were presented as mean±SDs for mother or father and male or female offspring separately. Also the mean±SDs of nutrient intakes of grandfathers or mothers and their grandson or daughter dyads were shown and the significant differences were estimated using paired t-test. To ensure the adequacy of the available sample size for comparing the paired means, the power of study was calculated; in 80% of matched pairs, the power of analysis was ≥80%. The correlation of energy and energy adjusted nutrient intakes of parents-offspring or grandparents-grandson or daughter dyads were estimated using partial correlation. Correlations were adjusted for parental and offspring age, body mass index (BMI) and physical activity. These analyses were performed based on living arrangement. Fisher's Z transformation test was applied to r-weighted by the sample size to ensure the correlations of dietary intakes between groups (two sets of familial dyads, outside or inside the family) were comparable.

Linear regression models were used to predict offspring dietary intakes by living arrangements (living with their parents, living independently with their parents). Main exposure was parent's dietary intake; this model was adjusted for parents' age, education, smoking and body mass index. Also linear regression models were used to predict grandson/daughter dietary intakes based on grandparents dietary intakes. Main exposure was grandparent's dietary intake; this model was adjusted for grandparents' age, education, smoking and body mass index.

To compare multiple tests, a false discovery rate (FDR) adjusted P value<0.2 was used and P<0.01 was considered to be significant based on <20 tests.

## Results

The characteristics of grandparents, parents and offspring by age (≥20 and <20), gender and living status were shown in Table 1. The mean age of grandfathers and grandmothers were 69.4±7.9 and 63.7±8.5 respectively, whose dietary data were dichotomized with their grandsons or granddaughters. The percentage of smokers among grand-fathers was 11.7%. There were 857 fathers (mean age: 55.4±11.1) and 1394 mothers (mean age: 50.1±11.4) whose dietary data were coupled with their boys or girls by living arrangements. The percentage of smokers among fathers and mothers were 21.6 and 2.7%. respectively. Of girls and boys (mean age: 28.8

**Table 1. Characteristics of 3 generations of study participants: Tehran lipid and glucose study.**

| Characteristics | Grand-father | Grand-mother | Father | Mother | Offspring (living with their parents)[a] | | | | Offspring (living independent of their parents)[b] | |
|---|---|---|---|---|---|---|---|---|---|---|
| | | | | | Boys | | Girls | | Men | Women |
| | | | | | ≥20 | <20 | ≥20 | <20 | | |
| **n** | 111 | 188 | 857 | 1394 | 513 | 355 | 515 | 361 | 469 | 552 |
| Age (years) | 69.4±7.9 [c] | 63.7±8.5 | 55.4±11.1 | 50.1±11.4 | 28.8±7.8 | 13.3±4.11 | 28.0±6.33 | 12.9±4.21 | 37.0±8.0 | 33.8±8.1 |
| Education >12 years (%) | 12.5 | 1.1 | 25.1 | 15.8 | 57.2 | 3.4 | 48.9 | 2.8 | 48.2 | 48.3 |
| Smokers (%) | 11.7 | 2.1 | 21.6 | 2.7 | 4.7 | 0 | 24.9 | 7.4 | 25.8 | 2.9 |
| BMI (Kg/m$^2$) | 26.8±3.8 | 31.0±4.8 | 27.5±4.4 | 29.9±5.0 | 24.3±4.6 | 21.4±5.8 | 25.6±4.4 | 20.6±5.0 | 27.9±4.5 | 27.0±5.0 |
| WC (cm) | 97.7±9.6 | 99.6±11.0 | 97.6±10.4 | 94.9±11.9 | 80.4±10.4 | 71.9±11.7 | 90.8±11.4 | 74.7±15.5 | 97.3±10.8 | 86.5±11.5 |
| Physical activity (MET/h/week) | 700±769 | 407±606 | 574±906 | 484±741 | 429±724 | 668±1026 | 916±1190 | 1356±1481 | 658±1268 | 488±749 |

[a] Offspring (living with their parents): Not married

[b] Offspring (living independent of their parents): Married.

[c] Data are means±SDs, unless otherwise listed.

[d] (The mean percentage of energy intake)

BMI: Body mass index; WC: Waist circumference.

±7.8 and 28.8±6.33, respectively) who lived with their parents, about 59% had ≥20 years old. The mean age of girls and boys aged <20 years were 13.3±4.11 and 12.9±4.21, respectively. The mean WC of girls aged ≥20 years were 80.4±10.4 cm. The physical activity of girls aged ≥20 and <20 were 429±724 and 668±1026 MET/h/week, respectively. In adult offspring (living independent of their parents) the mean physical activity of men and women were 658 ±1268 and 488±749 MET/h/week, respectively.

Total energy and nutrient intakes of three generations of participants (fathers-sons) by living status were shown in Table 2. The mean percentage of energy from total fat, SFA, trans-fatty acids, MUFA, PUFA and cholesterol (mg/day) of sons (living with their parents) were higher than their fathers. The mean percentage of energy from trans-fatty acids were >2% in both fathers and sons (living with their parents). The significant correlation of nutrient intakes in fathers and sons (living with their parents) pairs were observed for 9 nutrients.

There were no significant correlation of nutrient intakes in fathers and sons (not living with their parents). Also nutrient intakes of grandfather-grandson dyads were not correlated. The correlation of nutrient intakes including total, animal and vegetable protein, total fat, SFA and trans-fatty acids (percentage of energy), were stronger among father-son (living with their parents) than father-son (living independent of their parents) dyads.

For 7 nutrients, there were significant correlations between father-daughter (living with their parents) dyads (Table 3). The mean percentage of energy intake from total fat, trans-fatty acids, SFA, MUFA, PUFA, was higher in young daughters than their fathers. The fiber intake (gr/1000 kcal/day) of daughters (living independent of their parents) was significantly lower than their fathers, while the mean percentage of energy intakes from total fat, trans-fatty acids, SFA, MUFA and PUFA were higher in daughters (living independent of their parents) than their fathers. The correlation of fiber intake (gr/1000 kcal/day) between father-daughter (living with their parents) dyads was stronger than this correlation between father-daughter (living independent of their parents) and grandfather-granddaughter dyads. For all nutrients, there were no significant adjusted correlations between grandfathers and granddaughters dyads.

The mean difference and adjusted correlation of dietary intakes of mother-son were shown in Table 4 by living status. Correlations of nutrient intakes were significant for 13 nutrients in

**Table 2. Total energy and nutrient intakes of 3 generations of participants (fathers-sons) by living status.**

| | Fathers | Sons (living with their parents) | P[a] | r[b] | Fathers | Sons (living independent of their parents) | P[a] | r[b] | Grand-father | Grand-son | P[a] | r[b] |
|---|---|---|---|---|---|---|---|---|---|---|---|---|
| n (paired) | | 543 | | | | 260 | | | | 75 | | |
| Total energy (Kcal/day) | 2524 ±1176 | 2849±1138 | <0.001 | 0.08 | 2308 ±939 | 2677±1005 | <0.001 | -0.04 | 2390±962 | 3089 ±1559 | <0.001 | -0.04 |
| Carbohydrate | 61.1 ±6.35 | 58.2±6.55 | <0.001 | 0.11 | 59.6 ±6.48 | 59.2±6.28 | 0.34 | -0.005 | 61.1±7.5 | 56.9±6.6 | <0.001 | -0.05 |
| Starch [c] | 36.1 ±9.10 | 33.2±8.70 | <0.001 | 0.18 [f] | 31.1 ±9.47 | 34.0±9.36 | <0.001 | 0.06 | 33.9±10.4 | 32.2 ±10.5 | 0.25 | 0.19 |
| Non-starch [c] | 15.7 ±7.21 | 16.3±6.62 | 0.11 | 0.16 [f] | 17.8 ±7.52 | 15.7±6.93 | <0.001 | 0.10 | 17.4±7.2 | 15.4±6.5 | 0.06 | 0.04 |
| Protein | 14.6 ±2.34 | 14.5±2.46 | 0.66 | 0.15 [f] | 15.3 ±3.13 | 14.9±2.36 | 0.03 | -0.03 [g] | 15.1±5.8 | 14.6±2.4 | 0.19 | -0.13 [h] |
| Animal protein [c] | 7.91 ±3.78 | 8.39±4.03 | 0.01 | 0.26 [f] | 8.22 ±3.71 | 8.90±4.01 | 0.03 | -0.004 [g] | 7.59±4.16 | 8.89 ±4.04 | 0.02 | 0.09 |
| Vegetable protein [c] | 6.34 ±1.76 | 6.61±1.68 | <0.001 | 0.21 [f] | 5.77 ±1.83 | 5.81±1.75 | 0.80 | 0.05 [g] | 6.21±2.09 | 5.49 ±1.99 | 0.02 | 0.17 |
| Total fat [c] | 27.5 ±5.80 | 30.1±5.87 | <0.001 | 0.19 [f] | 29.1 ±6.04 | 28.8±5.83 | 0.45 | 0.002 [g] | 28.5±7.6 | 31.1±6.4 | 0.01 | 0.18 |
| SFA [c] | 8.97 ±2.92 | 10.2±2.82 | <0.001 | 0.12 | 9.18 ±3.08 | 9.46±2.77 | 0.21 | -0.06 [g] | 8.90±3.18 | 10.3±2.8 | 0.004 | 0.08 |
| Trans-fatty acids [c] | 2.06 ±1.53 | 2.49±1.38 | <0.001 | 0.16 [f] | 1.94 ±1.43 | 2.28±1.35 | 0.002 | 0.01 [g] | 2.02±1.42 | 2.51 ±1.17 | 0.02 | 0.04 |
| MUFA [c] | 9.08 ±2.17 | 9.86±2.46 | <0.001 | 0.17 [f] | 9.56 ±2.24 | 9.60±2.14 | 0.79 | 0.06 | 10.0±6.5 | 9.89 ±2.20 | 0.83 | 0.09 |
| PUFA [c] | 5.52 ±1.69 | 5.84±2.02 | 0.001 | 0.17 [f] | 5.91 ±1.78 | 5.77±1.78 | 0.31 | 0.06 | 5.91±2.68 | 5.84 ±1.80 | 0.84 | 0.09 |
| Fiber [d] | 9.64 ±3.22 | 8.62±2.64 | 0.004 | 0.08 | 11.3 ±3.56 | 8.97±2.72 | <0.001 | 0.05 | 10.6±3.18 | 8.62 ±3.02 | <0.001 | 0.03 |
| Cholesterol (mg/day) | 235 ±259 | 295±193 | <0.001 | 0.08 | 191±130 | 269±151 | <0.001 | -0.03 | 190±96.0 | 341±218 | <0.001 | -0.02 |
| Vitamin C [e] | 64.1 ±38.5 | 58.9±32.3 | 0.009 | 0.15 | 67.7 ±34.6 | 64.1±34.7 | 0.03 | 0.02 [g] | 62.9±31.0 | 57.0 ±35.8 | 0.009 | -0.02 |
| Calcium [e] | 555 ±177 | 552±179 | 0.72 | 0.01 | 593±185 | 556±198 | 0.03 | 0.06 | 619±184 | 561±209 | 0.03 | 0.04 |
| Iron [e] | 15.3 ±9.15 | 14.0±8.95 | 0.009 | 0.06 | 16.8 ±9.25 | 15.4±10.1 | 0.09 | -0.04 | 17.2±10.8 | 14.7 ±9.79 | 0.10 | -0.08 |
| Zinc [e] | 5.59 ±0.91 | 5.51±2.93 | 0.56 | 0.08 | 5.93 ±2.57 | 8.02±2.57 | 0.38 | 0.05 | 6.03±2.84 | 5.31 ±0.90 | 0.01 | 0.17 |
| Sodium (mg/day) | 1679 ±4640 | 1428±370 | 0.21 | 0.007 | 1666 ±540 | 1469±435 | <0.001 | 0.11 | 1484±458 | 1515 ±375 | 0.65 | -0.17 |
| Magnesium [e] | 204 ±37.1 | 185±34.8 | <0.001 | 0.09 | 213 ±38.6 | 198±53.4 | <0.001 | 0.02 | 218±39.4 | 186±39.4 | <0.001 | -0.004 |

[a] Paired t-test for the difference of energy and dietary intakes

[b] Partial correlation (adjusted for age, physical activity and body mass index).

[c] (% of energy intake), [d] gr/1000 Kcal/day, [e] mg/1000 Kcal/day, [f] P<0.01 (P<0.01 is considered to be significant based on false discovery rate.), [g] Significant difference between the r correlation of fathers-sons living with their parents/fathers-sons living independent of their parents using Fisher's Z transformation test; [h] Significant difference between the r correlation of fathers-sons living with their parents/grandfather-grandson.

SFA: Saturated fatty acid; MUFA: Mono-unsaturated fatty acid; PUFA: Poly unsaturated fatty acid.

mother-son (living with their parents) dyads. The percentage of energy from SFA, trans-fatty acids and cholesterol intakes were higher in sons (living with their parents) than their mothers, while MUFA and PUFA (percentage of energy), fiber gr/1000kcal/day, vitamin C, calcium,

**Table 3. Total energy and nutrient intakes of 3 generations of participants (fathers-daughters) by living status.**

| | Fathers | Daughters (living with their parents) | P [a] | r [b] | Fathers | Daughters (living independent of their parents) | P [a] | r [b] | Grand-fathers | Grand-daughters | P [a] | r [b] |
|---|---|---|---|---|---|---|---|---|---|---|---|---|
| **n** (paired) | 586 | | | | 302 | | | | 105 | | | |
| Total energy (Kcal/day) | 2567 ±1260 | 2515±1625 | 0.52 | 0.07 | 2456±900 | 2237±752 | <0.01 | 0.12 | 2487±8.35 | 2315±899 | 0.18 | 0.17 |
| Carbohydrate | 61.1±6.29 | 57.3±6.67 | <0.001 | 0.14 [f] | 61.1±6.35 | 58.2±6.55 | <0.001 | 0.17 [f] | 63.0±7.4 | 55.8±6.9 | <0.001 | 0.09 |
| Starch [c] | 35.8±9.45 | 30.4±8.63 | 0.001 | 0.06 | 36.1±9.10 | 33.2±8.76 | <0.001 | 0.07 | 34.4±10.9 | 30.3±9.9 | 0.02 | -0.04 |
| Non-starch [c] | 16.1±7.09 | 17.5±7.49 | 0.001 | 0.19 [f] | 15.7±7.21 | 16.3±6.62 | 0.11 | 0.11 | 18.4±8.7 | 16.6±7.1 | 0.14 | 0.22 |
| Protein | 14.6±2.14 | 14.2±2.43 | 0.003 | 0.21 [f] | 14.6±2.34 | 14.5±2.46 | 0.66 | 0.13 | 15.2±3.10 | 14.3±2.0 | 0.001 | 0.14 |
| Animal protein [c] | 8.03±4.10 | 8.21±3.84 | 0.44 | 0.09 | 7.91±3.78 | 8.39±4.03 | 0.01 | 0.16 [f] | 7.91±4.45 | 9.44±6.33 | 0.06 | 0.09 |
| Vegetable protein [c] | 6.31±1.79 | 5.14±1.65 | 0.001 | 0.08 | 6.34±1.76 | 5.60±1.68 | <0.001 | 0.08 | 6.23±2.16 | 5.05±1.90 | <0.001 | 0.02 |
| Total fat [c] | 27.6±5.67 | 31.6±6.04 | <0.001 | 0.20 [f] | 27.5±5.81 | 30.1±5.87 | <0.001 | 0.18 [f] | 26.5±6.89 | 32.6±6.85 | <0.001 | 0.05 |
| SFA [c] | 8.93±2.60 | 10.4±2.77 | <0.001 | 0.12 | 8.97±2.92 | 10.2±2.82 | <0.001 | 0.07 | 8.35±3.34 | 11.1±2.3 | <0.001 | 0.04 |
| Trans-fatty acids [c] | 2.03±1.26 | 2.63±1.46 | <0.001 | 0.12 | 2.06±1.53 | 2.49±1.37 | <0.001 | 0.13 | 1.70±1.15 | 2.48±1.22 | <0.001 | 0.24 |
| MUFA [c] | 9.08±2.05 | 10.5±2.54 | <0.001 | 0.21 [f] | 9.08±2.18 | 9.86±2.46 | <0.001 | 0.18 [f] | 9.55±6.70 | 10.7±2.55 | 0.18 | 0.07 |
| PUFA [c] | 5.54±1.63 | 6.33±2.04 | <0.001 | 0.16 [f] | 5.52±1.69 | 5.84±2.01 | 0.001 | 0.14 [f] | 5.45±1.97 | 6.39±2.25 | 0.004 | 0.33 [h] |
| Fiber [d] | 9.70±3.01 | 9.65±3.52 | 0.76 | 0.27 [f] | 9.64±3.22 | 8.62±2.64 | 0.004 | 0.11 [g] | 11.2±3.71 | 9.20±2.84 | <0.01 | 0.01 [h] |
| Cholesterol (mg/day) | 242±272 | 223±112 | 0.13 | 0.08 | 235±238 | 295±192 | <0.001 | 0.09 | 206±105 | 217±114 | 0.005 | 0.09 |
| Vitamin C [e] | 64.6±33.7 | 69.7±44.3 | 0.01 | 0.14 | 71.4±38.3 | 73.0±35.6 | 0.59 | 0.06 | 79.9±48.0 | 59.6±28.5 | 0.003 | 0.17 |
| Calcium [e] | 543±179 | 576±191 | 0.001 | 0.16 [f] | 565±191 | 595±182 | 0.04 | 0.04 | 613±209 | 597±185 | 0.63 | 0.27 |
| Iron [e] | 14.9±9.22 | 14.9±11.0 | 0.99 | 0.21 [f] | 16.1±10.4 | 14.7±8.70 | 0.07 | 0.05 | 18.8±11.9 | 13.9±8.22 | 0.005 | 0.12 |
| Zinc [e] | 5.64±2.0 | 5.58±4.64 | 0.76 | 0.62 | 5.56±0.94 | 5.41±0.86 | 0.03 | 0.04 | 6.17±3.33 | 5.27±0.76 | 0.02 | 0.28 |
| Sodium (mg/day) | 1718±481 | 1510±431 | 0.33 | 0.01 | 1625 ±1469 | 1735±2608 | 0.41 | -0.04 | 1424±394 | 1455±429 | 0.64 | 0.12 |
| Magnesium [e] | 204±43.4 | 187±37.8 | <0.001 | 0.13 | 208±49.5 | 191±35.0 | <0.01 | 0.08 | 222±42.1 | 183±31 | <0.001 | -0.04 |

[a] Paired t-test for the difference of energy and dietary intakes

[b] Partial correlation (adjusted for age, physical activity and body mass index).

[c] (% of energy intake), [d] gr/1000 Kcal/day, [e] mg/1000 Kcal/day, [f] P<0.01 (P<0.01 is considered to be significant based on false discovery rate.), [g] Significant difference between the r correlation of fathers-daughters living with their parents/fathers-daughters living independent of their parents using Fisher's Z transformation test; [h] Significant difference between the r correlation of fathers-daughters living with their parents/grandfather-granddaughter.

SFA: Saturated fatty acid; MUFA: Mono-unsaturated fatty acid; PUFA: Poly unsaturated fatty acid.

iron and magnesium mg/1000kcal/day intake of sons (living with their parents) were lower than their mothers. Only one correlation (magnesium intake) was significant for mother-son (living independent of their parents) dyads. Correlation coefficients for some nutrients were stronger for mother-son (living with their parents) dyads than mother-son (living independent of their parents) dyads. The fiber intake (gr/1000 kcal/day) of grandmothers-grandson dyads was correlated. Animal protein and total fat (percentage of energy) and cholesterol (mg/day) intakes of grandsons were higher than grandmothers.

For all nutrients except trans-fatty acids, there were significant correlations between dietary intakes of mother-daughter (living with their parents) dyads (Table 5). Fisher's Z transformation showed that the correlation of nutrient intakes between mother-daughter (living with their parents) were stronger than the correlation of nutrient intakes between mother-daughter (living independent of their parents) dyads.

Daughters had a higher intake of total fat, SFA, trans-fatty acids, MUFA, PUFA (percentage of energy) and cholesterol (mg/day) than their grandmothers. The fiber (gr/1000 Kcal/day),

**Table 4. Total energy and nutrient intakes of 3 generations of participants (mothers-sons) by living status.**

| | Mothers | Sons (living with their parents) | P [a] | r [b] | Mothers | Sons (living independent of their parents) | P [a] | r [b] | Grand-mothers | Grand-sons | P [a] | r [b] |
|---|---|---|---|---|---|---|---|---|---|---|---|---|
| **n** (paired) | | 746 | | | | 353 | | | | 164 | | |
| Total energy (Kcal/day) | 2372 ±1076 | 2879±1190 | <0.01 | 0.11 [f] | 2213 ±881 | 2728±1074 | <0.01 | -0.01 [g] | 2287±938 | 2996 ±1104 | <0.01 | -0.03 [h] |
| Carbohydrate | 58.3 ±6.84 | 58.3±6.52 | 0.90 | 0.18 [f] | 59.6 ±6.50 | 59.4±6.14 | 0.70 | -0.02 [g] | 59.4±7.0 | 57.8±6.4 | 0.05 | 0.07 |
| Starch [c] | 30.9 ±9.31 | 33.5±9.03 | <0.001 | 0.09 | 31.3±9.4 | 34.0±9.4 | <0.001 | 0.06 | 31.9±9.27 | 33.0 ±8.64 | 0.26 | 0.24 [h] |
| Non-starch [c] | 17.3 ±8.10 | 16.2±7.0 | 0.002 | 0.23 [f] | 17.9 ±7.56 | 15.8±6.99 | <0.001 | 0.10 [g] | 17.8±7.7 | 16.1±6.1 | 0.03 | 0.26 |
| Protein | 14.8 ±2.63 | 14.6±2.51 | 0.24 | 0.14 [f] | 15.4 ±3.19 | 14.9±2.38 | 0.02 | 0.02 [g] | 15.1±3.6 | 14.8 ±2.62 | 0.25 | 0.18 |
| Animal protein [c] | 8.27 ±4.45 | 8.43±4.09 | 0.41 | 0.22 [f] | 8.22 ±3.69 | 8.87±3.98 | 0.03 | -0.03 [g] | 7.75±3.88 | 8.87 ±4.24 | 0.009 | 0.25 |
| Vegetable protein [c] | 5.63 ±1.81 | 5.68±1.68 | 0.63 | 0.15 [f] | 5.76 ±1.82 | 5.81±1.75 | 0.67 | 0.07 | 5.90±1.90 | 5.54 ±1.57 | 0.08 | 0.08 |
| Total fat [c] | 30.4 ±6.66 | 30.0±5.87 | 0.19 | 0.15 [f] | 29.1 ±5.96 | 28.5±5.86 | 0.19 | 0.02 [g] | 29.2±6.40 | 30.1±5.9 | 0.23 | 0.06 |
| SFA [c] | 9.53 ±2.70 | 10.1±2.83 | <0.001 | 0.12 [f] | 9.17 ±3.07 | 10.8±25.0 | 0.23 | -0.02 [g] | 8.94±2.51 | 10.0 ±2.71 | <0.001 | 0.17 |
| Trans-fatty acids [c] | 2.17 ±1.60 | 2.43±1.30 | <0.001 | 0.22 [f] | 1.94 ±1.44 | 2.28±1.35 | 0.002 | 0.04 [g] | 2.03±1.36 | 2.34 ±1.17 | 0.03 | -0.03 [h] |
| MUFA [c] | 10.2 ±3.01 | 9.80±2.32 | 0.006 | 0.11 | 9.57 ±2.24 | 9.61±2.14 | 0.79 | 0.07 | 9.60±2.29 | 9.72 ±1.97 | 0.61 | 0.11 |
| PUFA [c] | 6.24 ±2.08 | 5.85±1.97 | <0.001 | 0.12 [f] | 5.91 ±1.84 | 5.76±1.79 | 0.27 | 0.08 | 5.99±1.87 | 5.79 ±1.55 | 0.26 | 0.01 |
| Fiber [d] | 10.8 ±3.46 | 8.75±2.76 | <0.001 | 0.18 [f] | 11.3 ±3.59 | 8.98±2.73 | <0.001 | 0.05 [g] | 11.1±3.57 | 8.78 ±2.79 | <0.001 | 0.35 [f, h] |
| Cholesterol (mg/day) | 206±116 | 299±193 | <0.001 | 0.08 | 191±130 | 269±151 | <0.001 | -0.003 | 188±144 | 330±232 | <0.001 | -0.08 [h] |
| Vitamin C [e] | 75.9 ±41.3 | 58.6±33.3 | <0.001 | 0.17 [f] | 79.5 ±38.4 | 61.9±33.9 | <0.001 | 0.08 | 77.1±40.8 | 60.3 ±33.8 | <0.001 | 0.17 |
| Calcium [e] | 601±215 | 548±176 | <0.001 | 0.07 | 649±220 | 559±193 | <0.001 | 0.11 | 641±225 | 554±191 | <0.001 | 0.24 [h] |
| Iron [e] | 16.6 ±11.7 | 13.7±8.29 | <0.001 | 0.10 | 18.8 ±12.3 | 15.5±10.3 | <0.001 | 0.03 [g] | 18.1±12.0 | 14.0 ±8.62 | <0.001 | 0.19 |
| Zinc [e] | 5.64 ±1.42 | 5.50±2.50 | 0.14 | 0.07 | 5.78 ±1.04 | 7.35±3.03 | 0.37 | 0.06 | 5.64±0.99 | 5.34 ±0.82 | 0.003 | 0.26 [h] |
| Sodium (mg/day) | 1599 ±425 | 1433±366 | <0.001 | 0.09 | 1667 ±540 | 1465±440 | <0.001 | 0.12 | 1651±545 | 1442 ±337 | <0.001 | 0.15 |
| Magnesium [e] | 205±43.9 | 187±34.7 | <0.001 | 0.16 [f] | 216±40.4 | 197±49.2 | <0.001 | 0.18 [f] | 213±44.0 | 185±33.4 | <0.001 | 0.14 |

[a] Paired t-test for the difference of energy and dietary intakes

[b] Partial correlation (adjusted for age, physical activity and body mass index).

[c] (% of energy intake)

[d] gr/1000 Kcal/day

[e] mg/1000 Kcal/day, [f] P<0.01 (P<0.01 is considered to be significant based on false discovery rate.), [g] Significant difference between the r correlation of mothers-sons living with their parents/mothers-sons living independent of their parents using Fisher's Z transformation test; [h] Significant difference between the r correlation of mothers-sons living with their parents/grandmother-grandson.

SFA: Saturated fatty acid; MUFA: Mono-unsaturated fatty acid; PUFA: Poly unsaturated fatty acid.

**Table 5. Total energy and nutrient intakes of 3 generations of participants (Mothers-daughters) by living status.**

| | Mothers | Daughters (living with their parents) | P [a] | r [b] | Mothers | Daughters (living independent of their parents) | P [a] | r [b] | Grand-mothers | Grand-daughters | P [a] | r [b] |
|---|---|---|---|---|---|---|---|---|---|---|---|---|
| **n** (paired) | 729 | | | | 346 | | | | 148 | | | |
| Total energy (Kcal/day) | 2378±1071 | 2501±1532 | 0.04 | 0.19 [f] | 2287±1021 | 2310±805 | 0.69 | 0.12 | 2226±922 | 2373±1769 | 0.34 | 0.14 |
| Carbohydrate | 58.2±7.26 | 57.0±6.86 | <0.001 | 0.36 [f] | 58.0±6.83 | 57.1±6.09 | 0.02 | 0.17 [fg] | 57.2±7.56 | 57.1±6.86 | 0.01 | 0.09 |
| Starch [c] | 30.5±9.45 | 30.8±8.57 | 0.56 | 0.34 [f] | 30.0±8.92 | 29.4±8.58 | 0.28 | 0.07 [g] | 31.8±9.25 | 30.4±9.0 | 0.18 | 0.25 [h] |
| Non-starch [c] | 17.6±8.66 | 17.1±7.21 | 0.11 | 0.26 [f] | 17.8±6.94 | 17.8±7.71 | 0.94 | 0.11 [g] | 17.4±7.22 | 17.5±7.11 | 0.89 | 0.28 |
| Protein | 14.4±2.45 | 14.3±2.38 | 0.10 | 0.35 [f] | 14.7±2.93 | 14.7±2.48 | 0.77 | 0.13 [g] | 15.4±3.8 | 14.2±1.9 | 0.001 | 0.02 [h] |
| Animal protein [c] | 8.24±4.95 | 8.34±4.55 | 0.65 | 0.31 [f] | 7.93±3.94 | 8.94±4.01 | 0.001 | 0.16 [fg] | 8.03±3.77 | 8.77±4.51 | 0.14 | 0.09 [h] |
| Vegetable protein [c] | 5.52±1.82 | 5.21±1.70 | <0.001 | 0.26 [f] | 5.47±1.71 | 5.13±1.66 | 0.005 | 0.08 [g] | 5.80±1.86 | 5.17±1.62 | 0.002 | 0.24 |
| Total fat [c] | 30.8±6.85 | 31.8±6.40 | 0.002 | 0.27 [f] | 30.8±6.62 | 31.3±5.74 | 0.26 | 0.18 [f] | 29.5±6.6 | 31.7±6.6 | 0.005 | 0.11 [h] |
| SFA [c] | 9.71±2.83 | 10.4±2.89 | <0.001 | 0.29 [f] | 9.57±3.10 | 10.1±2.56 | 0.009 | 0.07 [g] | 9.43±3.48 | 10.5±3.12 | 0.004 | 0.06 [h] |
| Trans-fatty acids [c] | 2.18±1.92 | 2.57±1.42 | <0.001 | 0.10 | 2.08±1.37 | 2.31±1.28 | 0.01 | 0.13 | 1.95±1.32 | 2.46±1.52 | 0.003 | 0.13 |
| MUFA [c] | 10.2±2.71 | 10.5±2.62 | 0.01 | 0.34 [f] | 10.1±2.48 | 10.4±2.27 | 0.07 | 0.18 [fg] | 9.66±2.47 | 10.4±2.67 | 0.004 | 0.34 [f] |
| PUFA [c] | 6.27±2.15 | 6.33±2.09 | 0.56 | 0.33 [f] | 6.27±1.88 | 6.17±1.87 | 0.43 | 0.14 [fg] | 5.98±2.09 | 6.24±2.17 | 0.23 | 0.24 |
| Fiber [d] | 10.8±3.50 | 9.51±3.31 | <0.001 | 0.30 [f] | 11.1±3.32 | 10.3±3.25 | <0.001 | 0.11 [g] | 10.8±3.12 | 9.76±2.99 | 0.008 | -0.001 [h] |
| Cholesterol (mg/day) | 204±97.2 | 222±110 | <0.001 | 0.29 [f] | 194±106 | 218±109 | 0.001 | 0.09 [g] | 187±143 | 211±114 | 0.09 | 0.04 [h] |
| Vitamin C [e] | 76.3±42.3 | 68.2±46.6 | <0.001 | 0.21 [f] | 79.7±39.5 | 73.8±36.0 | 0.02 | 0.08 [g] | 74.3±37.8 | 67.8±35.1 | 0.10 | 0.13 |
| Calcium [e] | 591±199 | 579±201 | 0.17 | 0.37 [f] | 622±223 | 598±192 | 0.08 | 0.11 [g] | 643±213 | 592±182 | 0.01 | 0.08 [h] |
| Iron [e] | 16.0±10.2 | 14.9±11.9 | 0.03 | 0.46 [f] | 17.2±11.9 | 15.0±9.23 | 0.004 | 0.02 [g] | 17.8±11.9 | 14.6±9.35 | 0.007 | -0.03 [h] |
| Zinc [e] | 5.52±1.28 | 5.63±4.50 | 0.53 | 0.25 [f] | 5.58±1.07 | 5.52±1.17 | 0.35 | 0.06 [g] | 5.76±1.25 | 5.40±0.95 | 0.005 | -0.11 [h] |
| Sodium (mg/day) | 1591±427 | 1515±411 | <0.001 | 0.17 [f] | 1625±1469 | 1735±2608 | 0.41 | -0.04 [g] | 1644±569 | 1540±445 | 0.09 | -0.14 [h] |
| Magnesium [e] | 199±40.3 | 188±40.3 | <0.001 | 0.32 [f] | 207±44.4 | 193±35.9 | <0.001 | 0.18 [fg] | 211±43.6 | 189±37.5 | <0.001 | -0.08 [h] |

[a] Paired t-test for the difference of energy and dietary intakes

[b] Partial correlation (adjusted for age, physical activity and body mass index).

[c] (% of energy intake)

[d] gr/1000 Kcal/day

[e] mg/1000 Kcal/day

[f] P<0.01 (P<0.01 is considered to be significant based on false discovery rate).

[g] Significant difference between the r correlation of mothers-daughters living with their parents/mothers-daughters living independent of their parents using Fisher's Z transformation test

[h] Significant difference between the r correlation of mothers-daughters living with their parents/grandmother-granddaughters.

SFA: Saturated fatty acid; MUFA: Mono-unsaturated fatty acid; PUFA: Poly unsaturated fatty acid.

calcium, magnesium, iron, zinc and vitamin C (mg/1000 Kcal/day) intake was lower in daughters than their grandmothers. Fisher's Z transformation showed that the correlation of some nutrient intakes between mother-daughter (living with their parents) were stronger than the correlation of nutrient intakes between grandmother-granddaughter dyads.

Linear regression analysis explaining parents-offspring dietary intakes resemblance by differences in living arrangements was shown in S1 Table. Parents' dietary intakes were positively associated with son/daughters' dietary intakes who lived with their parents; however, dietary intakes of males or females who lived independently of their parents did not be predicted by parents' dietary intake except for carbohydrate, protein, total fat, SFA, MUFA and PUFA. Linear regression analysis explaining grandparents-grandson/daughter dietary intakes resemblance was shown in S2 Tbale. Grandparents' dietary intakes were not associated with grandson/daughter dietary intakes except for protein (β = 0.09, P = 0.002) and PUFA (β = 0.12, P = 0.008) intakes.

## Discussion

The present study suggests similarities between the nutrient intakes of parents and their children and adolescents who lived with them; the highest correlation was seen for mother-daughter nutrient intakes. Also the strength and significance of these associations were higher than relationships between nutrient intakes of parent-adult offspring who married and lived independently from their parents. This study was further demonstrated that there were weak to no correlations between the young offspring-grandparents dyads. Overall total fat, SFA, trans-fatty acids and cholesterol intakes of young offspring were higher than their parents, while dietary fiber intake of parents were higher than their offspring.

To our knowledge, few studies in developing countries examined the resemblance of nutrient intakes of parent- young or adult offspring pairs by sex and living arrangements in three generations [7,14,25].

Similarities were stronger for nutrient intakes of parent-young offspring dyads that lived with their parents than their peers not living at home with their parents, which was similar to previous study [14]. The nutrient intakes of married adult offspring are not affected by their parents. The socioeconomic and environmental factors may partially induce their food preferences. New food production and advertisement, local food environment and peer influence may affect the dietary intake of adult offspring [7]. This correlation for grandparents and their granddaughters or sons were diminished, which suggest that the influence of parental nutrient intakes on their offspring is likely to be disappeared [7,15]. The previous study reported that the presence of grandparents in the family structure had been related to unhealthy dietary intakes in children, including higher consumption of unhealthy snacks and larger proportion of meals, since grandparents believe that the heavier children are healthier or children who eat more will grow taller [1].

The similarity of nutrient intakes of parent and young unmarried offspring who lived with their parents may be due to common environmental variances like eating more meals at home with each other [6,8,26]. The number of family meals was positively associated with the consumption of healthier foods. Sharing meals together had been related to healthful dietary pattern and decreasing intakes of fast foods and sweetened drinks. Also factors such as parental authority, role modelling and parental control over access and availability of foods may influence the dietary intake of young offspring [1,27].

The percentage of energy intakes from total fat, SFA, trans-fatty acids and cholesterol intake of young offspring were higher than their parents, because adolescents may consume more snack foods and eat with their peers at school or restaurants [2]. Television viewing and

advertising products of high sugar and sodium such as fast foods, sweets and unhealthy snacks had been shown to contribute the overconsumption of high fat and high sugar foods [2]; this may explain dissimilarity in fat intakes among parents and their young offspring. Moreover, overall correlations of nutrient intakes between parent-young offspring were weak in a previous meta-analysis [7,14], which was in accordance with our findings.

Of parent-offspring dyads correlation of nutrient intakes, the strongest positive correlation was seen for mother-daughter pairs. Previous studies reported strong positive correlation between the diets of parents and their daughters. Most studies integrated mother and father data into parents' dietary intake, so the idea of which parent may have greater relationship with the child's or adolescent's dietary intake was limited [28]. Moreover previous studies found that higher similarities of dietary intakes between mother-daughters than mother-sons while other studies did not report such dissimilarity and found that parental dietary intake affect the diet of children regardless of age and gender [8,29–31].

Previous studies reported stronger mother–child dietary similarities which is consistent with our findings [9,32,33]. The stronger correlation of mother-young offspring dietary intakes may be due to reporting bias because mothers may be more responding on behalf of their children or adolescents, which may falsely show the higher correlation and dietary resemblance [8]. Moreover, mothers may donate more time on cooking and preparing meals compared to fathers [6].

It seems that parent-offspring similarity of dietary intakes may be stronger for healthy foods rather than unhealthy food consumption [6,8]. Also previous studies reported that mothers have greater impact on child's total energy intake and nutrient-dense foods [6,8,9,32]; however this result does not accord with our findings in which nutrients were correlated in the range of 0.1–0.37.

This study has several limitations which should be considered; this is a cross-sectional study, thus causal relationship cannot be judged. The mother can remember of what her children or adolescents consumed at home and do not know exactly of what they eat outside the home; this may result in reporting errors. We did not have data about the number of shared meals of parent-offspring dyads to be adjusted. Also the number of meals at restaurants or outside the home was not specified. The TLGS is not long enough to estimate the existence of male or female line transgenerational effects; there were no earlier observations linking the paternal or maternal dietary intakes or behaviors during the childhood or adolescence, linking to offspring or grandchild dietary intakes or behaviors.

Strengths of our study include assessing dietary intake using FFQ, which shows the usual estimate of dietary consumption of parents-offspring pairs. Also this study provides parent-offspring dietary intakes by sex separately, which provides further perception in this relationship. Furthermore, we compared energy adjusted food group intakes of parents-offspring to better compare usual nutrient intakes of children and their parents in different age groups. Nutrients were adjusted for energy intake (percentage of energy or per/1000 kcal of energy intake).

Conclusion: The nutrient intakes of adult married offspring were not affected by their parents; this correlation of younger and older generations disappeared. There were weak to moderate correlation between nutrient intakes of parent-offspring dyads that lived with their parents. The resemblance was higher for mother-daughter than mother-son or for father-son than father-daughter. Overall, total fat, SFA, trans-fatty acids, cholesterol and calcium intakes of young offspring were higher than their parents, while dietary fiber intake of parents were higher than their offspring. Further studies with higher quality longitudinal designs are needed to confirm intergenerational dietary effects.

## Supporting information

**S1 Table. Linear regression model for predicting offspring dietary intakes by differences in living arrangements.**
(DOCX)

**S2 Table. Linear regression model for predicting grandson/daughter dietary intakes.**
(DOCX)

## Acknowledgments

The authors would like to thank the participants and the TLGS personnel for their collaboration.

## Author Contributions

**Conceptualization:** Parvin Mirmiran, Firoozeh Hosseini-Esfahani, Mahdi Akbarzadeh, Maryam S Daneshpour, Fereidoun Azizi.

**Data curation:** Firoozeh Hosseini-Esfahani.

**Formal analysis:** Asiyeh Sadat Zahedi, Glareh Koochakpour, Mahdi Akbarzadeh.

**Investigation:** Maryam S Daneshpour.

**Methodology:** Asiyeh Sadat Zahedi, Glareh Koochakpour, Firoozeh Hosseini-Esfahani, Mahdi Akbarzadeh, Maryam S Daneshpour, Fereidoun Azizi.

**Supervision:** Maryam S Daneshpour, Fereidoun Azizi.

**Writing – original draft:** Asiyeh Sadat Zahedi, Glareh Koochakpour, Firoozeh Hosseini-Esfahani, Mahdi Akbarzadeh.

**Writing – review & editing:** Parvin Mirmiran, Firoozeh Hosseini-Esfahani, Maryam S Daneshpour, Fereidoun Azizi.

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
