## [Decision Letter · Decision Letter 0]

8 Nov 2021

PONE-D-21-23425Resemblance of nutrient intakes in three generations of parent-offspring pairs: Tehran lipid and Glucose StudyPLOS ONE

Dear Dr. Hosseini-Esfahani,

Thank you for submitting your manuscript to PLOS ONE. After careful consideration, we feel that it has merit but does not fully meet PLOS ONE’s publication criteria as it currently stands. Therefore, we invite you to submit a revised version of the manuscript that addresses the points raised during the review process. Specifically the major limitations of the study pointed out by the reviewers along with their implications have to be discussed and the manuscript should be thoroughly edited. Statistical analysis related comments raised also need to be addressed as well.

We look forward to receiving your revised manuscript.

Kind regards,

Samson Gebremedhin, PhD

Academic Editor

PLOS ONE

Journal Requirements:

Additional Editor Comments (if provided):

As the reviewers pointed out, ideally the research question should have been answered through a longitudinal study. Furthermore, parents-offspring dyads nutrient intake should have been compared at the same age so that age-related differences in dietary intake can be offset. This is a major limitation that has to be acknowledged and be discussed in depth.

The manuscript needs to be thoroughly edited,

Abstract:

• The abstract has to be structured according to the journal guideline,

• “Pairwise partial t-test” do you mean paired t-test and partial correlation?

• “The correlation in fathers-sons and father-daughter (living with their parents) pairs 40 were observed for 13 and 11 nutrients” why same set of nutrients have not been compared?

• The results sub-section is difficult to understand and has to be revisited.

Background

• Line 65-66: the sentence “Some food groups such as fruit and vegetable had stronger correlation between mothers and children than other unhealthy foods” is confusing. It gives a negative connotation that fruits and vegetables are unhealthy foods.

Methods

• Please comment on the adequacy of the available sample size for comparing the paired means.

• Line 131-134: why you were interested in these nutrients? Why not in others like vitamin A and B vitamins?

Results and discussion

The increased possibility of type I error from repeated statistical testing has to be discussed.

Reviewers' comments:

Reviewer's Responses to Questions

**Comments to the Author**

1. Is the manuscript technically sound, and do the data support the conclusions?

Reviewer #1: No

Reviewer #2: Partly

2. Has the statistical analysis been performed appropriately and rigorously? 

Reviewer #1: No

Reviewer #2: No

3. Have the authors made all data underlying the findings in their manuscript fully available?

Reviewer #1: No

Reviewer #2: Yes

4. Is the manuscript presented in an intelligible fashion and written in standard English?

Reviewer #1: Yes

Reviewer #2: Yes

5. Review Comments to the Author

Reviewer #1: The study by Mirmiran et al. presents cross-sectional findings on the correlation of nutrient intakes between parent-child or grandparent-child dyads according to the child’s age and living arrangements. While it is interesting and relevant to quantify how much of our eating habits have been influenced through familial generations, the study design and analysis lacks the rigor to support the conclusions of the paper. More details follow:

1) The study could be strengthened by having longitudinal data. It would be interesting to compare dietary intakes of different generations at similar ages, or if that is not possible, to have a better estimate of usual intake of individuals across time. Lines 89-95 note that the study that this population is from had data collected at multiple time points – is it possible to leverage that longitudinal data?

2) A review by a statistician could provide insights into the best analytical approach. As is, I think the statistical approach lacks the rigor to fully test the objective of the paper. There is no comparison upon which to judge if the correlations between familial dyads are stronger or weaker than they would be with others the same age and/or gender outside of the family. Perhaps a hierarchical statistical model could be a useful approach to determine within vs between family correlations within the model.

3) The last sentence of the conclusion is not supported by the data. This study did not examine the healthfulness of the study participants’ diets.

4) The tables do not include units for the nutrients. These should be added.

5) The authors did not provide information on where the study data could be found.

6) The manuscript should be reviewed to correct some minor English errors throughout.

Reviewer #2: Overall this is a very important work and addresses a research question that is of high interest in the field. However, there are things that the authors can address and the soundess and value of the mansucript can increase significantly. I have some general and specific comments.

General comments:

I would strongly recommend to design Regression models for each nutrient adjusted for relevant and available covariates to further provide info on the influence of (1) living or not with parents on intake, (2) the specific relationship (e.g. grandmother/father- granddaughter/son or father/mother-son/daughter, in addition to correlation analysis that you have conducted. Providing the betas and bolding them to indicate significant ones, will add significant value to the manuscript and will transform its nature to a more in-depth and valuable analytical work. In addition you do not need to show all nutrients in the tables and you can provide the full tables and the full regression models in a supplementary file for the interested reader.

In this line of logic and based on the outcome of the regression analyses you can than discuss the potential genetic influence or presence of genomic imprinting (you can refer to this study to create an idea of what I mean: https://www.nature.com/articles/5201538)

Specific comments:

I have noticed many semantic, gramatical and ligcal typos which I would like to address so the qaulity of the work can be further improved:

Line 25: since you are not analyzing 'dietary habits', but nutrient intakes, consider replacing the word 'dietary habits' with 'dietary patterns'. This will create a consistency from the begining to thend of the mansucript.

Line 55: In line with my previous remark, consider intead of the word 'eating habits' the word 'nutrition', to be consistent and focused on your manuscript throghout the paper.

Line58: a hyphen is neeeded between diet and related

Line 70: after the coma and before the word they insert 'as'

Line 71: decreases not decrease

Line 72: number not numbers. On the same line you say that number of shared meals is less. Two things: (1) I would recommend to change the language to 'number of shared meals is usually less compared to.../or decreasing' (if there is a trend in the literature you are citing), after the reference 12 use 'but' to make the contrast and give the feeling to the reader to appreciate the research question you addressing from this point.

Lines 73-76: 'These finding raise this question...' It is a sentence that is starting weirdely and not flowing from the previous in logical terms. The entire sentence makes sense, however it is long and it mixes a lot of concepts. Thus, I recommend spliting it and modifiyng it in this way: 'Whether these dietary patterns, acquired from the family during childhood or adolescence, tend to track into adulthood after marriage and/or forming a separate family, remains a subject of specualtion. It is also unknown if such pattrerns persist or are maintained through multiple generations.' This way, the reader starts to get a clearer idea of the value of your work.

Line 77: After 'Most studies' insert 'have'

Line 78: Same issue here, after 'few studies' insert 'have'

Line 80: As far as understand, the authors have thought of a very niche topic and it is their merit to emphasize this. Therefore in this line I recommend modifyng the begining of the sientence like this: 'Moreover, theres is little or no evidence on the similarity/resemblance/relationship/association (choose one option that you consider most relevant) ....'

Line 82-83: Reove 'Also', and I recommend 'In addition, ...' and I recommend to split this sentence in two, with the second sentence starting after the coma (word literature). 'With that in mind, the aim of this study is to investigate ...' In this way you provide a a very clear idea of the value of your work.

Lines 85-86: This last sentence is too general, perhaps, provide a more concise potential contribution related to your topic: e.g. Findings can help elucidate intergenerational influences on dietary patterns or inform nutrition interventions targeting multigenerational households/extended families. I think in this way you become less generic and more precize.

Lines 89-90: insert hyphen between family and based.

Lines 101-102: extend explanation of these two sentences as they are a very important part of your work.

Line 113: inerviews (plural)

Tables: Adjust all nutrients when making the analysis, because I see that some nutrients are experessed as g/1000 kcal or %E, but some not (e.g. Table2, From cholesterol to magnesium). This is important as it may confound your statitical inferences.

6. PLOS authors have the option to publish the peer review history of their article (what does this mean?). If published, this will include your full peer review and any attached files.

Reviewer #1: No

Reviewer #2: **Yes: **Dr. Erand Llanaj, Epidemiologist at Public Health Research Group of the Eötvös Loránd Research Network

---

## [Author Response · Author response to Decision Letter 0]

13 Feb 2022

PONE-D-21-23425

Resemblance of nutrient intakes in three generations of parent-offspring pairs: Tehran lipid and Glucose Study

PLOS ONE

Editor Comments (if provided):

As the reviewers pointed out, ideally the research question should have been answered through a longitudinal study. Furthermore, parents-offspring dyads nutrient intake should have been compared at the same age so that age-related differences in dietary intake can be offset. This is a major limitation that has to be acknowledged and be discussed in depth.

Reply: Corrected, discussion, page 15, lines 303-306 and 310-312.

The TLGS is not long enough to estimate the existence of male or female line transgenerational effects; there were no earlier observations linking the paternal or maternal dietary intakes or behaviors during the childhood or adolescence, linking to offspring or grandchild dietary intakes or behaviors.

Furthermore, we compared energy adjusted food group intakes of parents-offspring to better compare usual nutrient intakes of children and their parents in different age groups. Nutrients were adjusted for energy intake (percentage of energy or per/1000 kcal of energy intake).

The manuscript needs to be thoroughly edited,

Reply: Agreed and Corrected.

Abstract:

• The abstract has to be structured according to the journal guideline,

Agreed and corrected.

• “Pairwise partial t-test” do you mean paired t-test and partial correlation?

Reply: Agreed and corrected, line 32.

• “The correlation in fathers-sons and father-daughter (living with their parents) pairs 40 were observed for 13 and 11 nutrients” why same set of nutrients have not been compared?

Reply: Corrected. Lines 36-37.

The same set of nutrients were compared (Tables 2 and 3); however only the number of significant correlations of nutrients were stated.

The significant correlation in fathers-sons and father-daughter (living with their parents) pairs were observed for 9 and 7 nutrients, respectively. 

• The results sub-section is difficult to understand and has to be revisited.

Reply: Agreed and corrected. Lines 175-240.

Background

• Line 65-66: the sentence “Some food groups such as fruit and vegetable had stronger correlation between mothers and children than other unhealthy foods” is confusing. It gives a negative connotation that fruits and vegetables are unhealthy foods.

Reply: Agreed and corrected. Lines 60-61.

Some healthy food groups such as fruit and vegetable had stronger correlation between mothers and children than unhealthy foods

Methods

• Please comment on the adequacy of the available sample size for comparing the paired means.

Reply: Agreed and corrected. Lines 158-160.

To ensure the adequacy of the available sample size for comparing the paired means, the power of study was calculated; for 80% of matched pairs, the power of analysis was ≥80%. 

• Line 131-134: why you were interested in these nutrients? Why not in others like vitamin A and B vitamins?

Reply: Agreed and corrected, lines 134-135.

The selection of nutrients was based on dietary guidelines; these nutrients were more discussed in dietary guidelines.

Results and discussion

The increased possibility of type I error from repeated statistical testing has to be discussed.

Reply: Agreed and corrected, page 9, lines 172-173 and tables 2-5 was corrected. 

To compare multiple tests, a false discovery rate (FDR) adjusted P value<0.2 was used and P<0.01 was considered to be significant based on <20 tests.

 

Reviewers' comments:

Reviewer's Responses to Questions

Comments to the Author

1. Is the manuscript technically sound, and do the data support the conclusions?

Reviewer #1: No

Reviewer #2: Partly

2. Has the statistical analysis been performed appropriately and rigorously?

Reviewer #1: No

Reviewer #2: No

3. Have the authors made all data underlying the findings in their manuscript fully available?

Reviewer #1: No

Reviewer #2: Yes

4. Is the manuscript presented in an intelligible fashion and written in standard English?

Reviewer #1: Yes

Reviewer #2: Yes

5. Review Comments to the Author

 

Reviewer #1: The study by Mirmiran et al. presents cross-sectional findings on the correlation of nutrient intakes between parent-child or grandparent-child dyads according to the child’s age and living arrangements. While it is interesting and relevant to quantify how much of our eating habits have been influenced through familial generations, the study design and analysis lacks the rigor to support the conclusions of the paper. More details follow:

1) The study could be strengthened by having longitudinal data. It would be interesting to compare dietary intakes of different generations at similar ages, or if that is not possible, to have a better estimate of usual intake of individuals across time. Lines 89-95 note that the study that this population is from had data collected at multiple time points – is it possible to leverage that longitudinal data?

Reply: Corrected as you mentioned. Page 15, lines 303-316.

The TLGS is not long enough to estimate the existence of male or female line transgenerational effects; there were no earlier observations linking the paternal or maternal dietary intakes or behaviors during the childhood or adolescence, linking to offspring or grandchild dietary intakes or behaviors.

2) A review by a statistician could provide insights into the best analytical approach. As is, I think the statistical approach lacks the rigor to fully test the objective of the paper. There is no comparison upon which to judge if the correlations between familial dyads are stronger or weaker than they would be with others the same age and/or gender outside of the family. Perhaps a hierarchical statistical model could be a useful approach to determine within vs between family correlations within the model.

Reply: Corrected as you mentioned. Pages 8-9, lines 163-171. Tables 2-5 and tables supplementary 1-2. Results, lines 196-198; 205-207; 216-218; 222-225; 229-231; 232-240.

Statistical methods: 

Fisher’s Z transformation test was applied to r-weighted by the sample size to ensure the correlations of dietary intakes between groups (two sets of familial dyads, outside or inside the family) were comparable. 

Linear regression models were used to predict offspring dietary intakes by living arrangements (living with their parents, living independently with their parents). Main exposure was parent’s dietary intake; this model was adjusted for parents’ age, education, smoking and body mass index. Also linear regression models were used to predict grandson/daughter dietary intakes based on grandparents dietary intakes. Main exposure was grandparent’s dietary intake; this model was adjusted for grandparents’ age, education, smoking and body mass index.

Results: 

The correlation of nutrient intakes including total, animal and vegetable protein, total fat, SFA and trans-fatty acids (percentage of energy), were stronger among father-son (living with their parents) than father-son (living independent of their parents) dyads.

The correlation of fiber intake (gr/1000 kcal/day) between father-daughter (living with their parents) dyads was stronger than this correlation between father-daughter (living independent of their parents) and grandfather-granddaughter dyads.

Correlation coefficients for some nutrients were stronger for mother-son (living with their parents) dyads than mother-son (living independent of their parents) dyads.

Fisher’s Z transformation showed that the correlation of nutrient intakes between mother-daughter (living with their parents) were stronger than the correlation of nutrient intakes between mother-daughter (living independent of their parents) dyads.

Fisher’s Z transformation showed that the correlation of some nutrient intakes between mother-daughter (living with their parents) were stronger than the correlation of nutrient intakes between grandmother-granddaughter dyads.

Linear regression analysis explaining parents-offspring dietary intakes resemblance by differences in living arrangements was shown in supplementary table 1. Parents’ dietary intakes were positively associated with son/daughters’ dietary intakes who lived with their parents; however, dietary intakes of males or females who lived independently of their parents did not be predicted by parents’ dietary intake except for carbohydrate, protein, total fat, SFA, MUFA and PUFA. Linear regression analysis explaining grandparents-grandson/daughter dietary intakes resemblance was shown in supplementary table 2. Grandparents’ dietary intakes were not associated with grandson/daughter dietary intakes except for protein (β=0.09, P=0.002) and PUFA (β=0.12, P=0.008) intakes.

3) The last sentence of the conclusion is not supported by the data. This study did not examine the healthfulness of the study participants’ diets.

Reply: Agreed and corrected. 

This sentence was removed.

4) The tables do not include units for the nutrients. These should be added.

Agreed and corrected. The units was added (Tables 2-5).

5) The authors did not provide information on where the study data could be found.

Reply: Page 5, line 87.

Residents of district 13 Tehran, the capital of Iran

6) The manuscript should be reviewed to correct some minor English errors throughout.

Agreed and corrected.

 

Reviewer #2: Overall this is a very important work and addresses a research question that is of high interest in the field. However, there are things that the authors can address and the soundess and value of the manuscript can increase significantly. I have some general and specific comments.

General comments:

I would strongly recommend to design Regression models for each nutrient adjusted for relevant and available covariates to further provide info on the influence of (1) living or not with parents on intake, (2) the specific relationship (e.g. grandmother/father- granddaughter/son or father/mother-son/daughter, in addition to correlation analysis that you have conducted. Providing the betas and bolding them to indicate significant ones, will add significant value to the manuscript and will transform its nature to a more in-depth and valuable analytical work. In addition you do not need to show all nutrients in the tables and you can provide the full tables and the full regression models in a supplementary file for the interested reader.

Reply: Agreed and corrected. 

Statistical methods: Page 9, lines 166-171.

Linear regression models were used to predict offspring dietary intakes by living arrangements (living with their parents, living independently with their parents). Main exposure was parent’s dietary intake; this model was adjusted for parents’ age, education, smoking and body mass index. Also linear regression models were used to predict grandson/daughter dietary intakes based on grandparents dietary intakes. Main exposure was grandparent’s dietary intake; this model was adjusted for grandparents’ age, education, smoking and body mass index.

Results: Page 12, lines 232-240.

Linear regression analysis explaining parents-offspring dietary intakes resemblance by differences in living arrangements was shown in supplementary table 1. Parents’ dietary intakes were positively associated with son/daughters’ dietary intakes who lived with their parents; however, dietary intakes of males or females who lived independently of their parents did not be predicted by parents’ dietary intake except for carbohydrate, protein, total fat, SFA, MUFA and PUFA. Linear regression analysis explaining grandparents-grandson/daughter dietary intakes resemblance was shown in supplementary table 2. Grandparents’ dietary intakes were not associated with grandson/daughter dietary intakes except for protein (β=0.09, P=0.002) and PUFA (β=0.12, P=0.008) intakes.

In this line of logic and based on the outcome of the regression analyses you can than discuss the potential genetic influence or presence of genomic imprinting (you can refer to this study to create an idea of what I mean: https://www.nature.com/articles/5201538)

Reply: Page 15, lines 303-306.

The TLGS is not long enough to estimate the existence of male or female line transgenerational effects; there were no earlier observations linking the paternal or maternal dietary intakes or behaviors during the childhood or adolescence, linking to offspring or grandchild dietary intakes or behaviors.

Specific comments:

I have noticed many semantic, gramatical and ligcal typos which I would like to address so the qaulity of the work can be further improved:

Line 25: since you are not analyzing 'dietary habits', but nutrient intakes, consider replacing the word 'dietary habits' with 'dietary patterns'. This will create a consistency from the begining to thend of the mansucript.

Reply: Corrected. Line 25.

Line 55: In line with my previous remark, consider intead of the word 'eating habits' the word 'nutrition', to be consistent and focused on your manuscript throghout the paper.

Reply: Agreed and corrected in all of the manuscript.

Line58: a hyphen is neeeded between diet and related

Reply: Corrected, line 53.

Line 70: after the coma and before the word they insert 'as'

Reply: Corrected, line 65.

Line 71: decreases not decrease

Reply: Corrected, line 66.

Line 72: number not numbers. On the same line you say that number of shared meals is less. Two things: (1) I would recommend to change the language to 'number of shared meals is usually less compared to.../or decreasing' (if there is a trend in the literature you are citing), after the reference 12 use 'but' to make the contrast and give the feeling to the reader to appreciate the research question you addressing from this point.

Reply: Corrected, line 68.

Lines 73-76: 'These finding raise this question...' It is a sentence that is starting weirdely and not flowing from the previous in logical terms. The entire sentence makes sense, however it is long and it mixes a lot of concepts. Thus, I recommend spliting it and modifiyng it in this way: 'Whether these dietary patterns, acquired from the family during childhood or adolescence, tend to track into adulthood after marriage and/or forming a separate family, remains a subject of specualtion. It is also unknown if such pattrerns persist or are maintained through multiple generations.' This way, the reader starts to get a clearer idea of the value of your work.

Reply: Agreed as you mentioned, lines 68-71.

These finding raise this question of whether these dietary intakes conformed in the family for children or adolescents tend to track into adulthood when they marry or form a separate family. It is also unknown if such patterns persist or maintained through multiple generations.

Line 77: After 'Most studies' insert 'have'

Reply:: Agreed as you mentioned, line 72.

Line 78: Same issue here, after 'few studies' insert 'have'

Reply: Agreed as you mentioned, line 73.

Line 80: As far as understand, the authors have thought of a very niche topic and it is their merit to emphasize this. Therefore in this line I recommend modifyng the begining of the sientence like this: 'Moreover, theres is little or no evidence on the similarity/resemblance/relationship/association (choose one option that you consider most relevant) ....'

Reply: Agreed as you mentioned, lines 75-76.

Line 82-83: Reove 'Also', and I recommend 'In addition, ...' and I recommend to split this sentence in two, with the second sentence starting after the coma (word literature). 'With that in mind, the aim of this study is to investigate ...' In this way you provide a a very clear idea of the value of your work.

Reply: Agreed as you mentioned, lines 77-82.

Lines 85-86: This last sentence is too general, perhaps, provide a more concise potential contribution related to your topic: e.g. Findings can help elucidate intergenerational influences on dietary patterns or inform nutrition interventions targeting multigenerational households/extended families. I think in this way you become less generic and more precize.

Reply: Agreed as you mentioned, lines 80-82.

Lines 89-90: insert hyphen between family and based.

Reply: Agreed as you mentioned, lines 85-86.

Lines 101-102: extend explanation of these two sentences as they are a very important part of your work.

Reply: Agreed as you mentioned, lines 94-98.

Among them, 1286 families (4685 subjects), who had at least two members of the family with complete data were entered as the population in the current cross-sectional study. These two members include parental (father or mother) and their female or male- children or adult offspring in two generations. In addition, data of parents with their young or adult offspring were paired based on living status. Also, data of grandparents and their grandson or daughter were coupled.

Line 113: inerviews (plural)

Reply: Agreed as you mentioned, line 109.

Tables: Adjust all nutrients when making the analysis, because I see that some nutrients are experessed as g/1000 kcal or %E, but some not (e.g. Table2, from cholesterol to magnesium). This is important as it may confound your statistical inferences.

Reply: Corrected as you mentioned. Lines 126-135 and tables 2-5.

To better compare usual nutrient intakes of children and their parents in two different age groups, nutrients were adjusted for energy intake (percentage of energy or per/1000 kcal of energy intake); e.g. including carbohydrate, starch and non-starch carbohydrate, protein, vegetable and animal protein, total fat, saturated fatty acid (SFA), mono-unsaturated fatty acid (MUFA), poly-unsaturated fatty acid (PUFA), trans-fatty acids (as percentage of energy), fiber (gr/1000 kcal/day), cholesterol (mg/day), sodium (mg/day), calcium, vitamin C, iron, zinc, and magnesium (as mg/1000 kcal/day). The recommended intakes of sodium and cholesterol are similar for all age groups. The selection of nutrients was based on dietary guidelines; these nutrients were more discussed in dietary guidelines.

---

## [Decision Letter · Decision Letter 1]

31 Mar 2022

Resemblance of nutrient intakes in three generations of parent-offspring pairs: Tehran lipid and Glucose Study

PONE-D-21-23425R1

Dear Dr. Hosseini-Esfahani,

We’re pleased to inform you that your manuscript has been judged scientifically suitable for publication and will be formally accepted for publication once it meets all outstanding technical requirements.

Kind regards,

Samson Gebremedhin, PhD

Academic Editor

PLOS ONE

Additional Editor Comments (optional):

Please address the minor comments raised by Reviewer II

Reviewers' comments:

Reviewer's Responses to Questions

**Comments to the Author**

1. If the authors have adequately addressed your comments raised in a previous round of review and you feel that this manuscript is now acceptable for publication, you may indicate that here to bypass the “Comments to the Author” section, enter your conflict of interest statement in the “Confidential to Editor” section, and submit your "Accept" recommendation.

Reviewer #2: All comments have been addressed

2. Is the manuscript technically sound, and do the data support the conclusions?

Reviewer #2: Yes

3. Has the statistical analysis been performed appropriately and rigorously? 

Reviewer #2: Yes

4. Have the authors made all data underlying the findings in their manuscript fully available?

Reviewer #2: Yes

5. Is the manuscript presented in an intelligible fashion and written in standard English?

Reviewer #2: Yes

6. Review Comments to the Author

Reviewer #2: The authors have followed the recommendations and improved the quality of the manuscript substantially. I feel that they have addressed my comments and remarks. I noticed three simple things that need modification:

(1) Line 25 change 'nutrient intakes' to 'nutrient intake patterns', as the sentence is more powerful and has more meaning this way.

(2) Line 44 and 314 that you say 'was disappeared', please remove 'was'.

(3) In your conclusions add a sentence indicating that your conclusions are incomplete and that further higher-quality studies with longitudinal designs are needed to confirm intergenerational dietary effects.

On a personal note to authors: it would be interesting if you consider investigating this line of study, for your next paper (same topic and data) try to focus on nutrient patterns and/or indicators to see intergenerational fidelity, as well as adherence to healthy and/or sustainable patterns and resemblance between generations.

7. PLOS authors have the option to publish the peer review history of their article (what does this mean?). If published, this will include your full peer review and any attached files.

Reviewer #2: **Yes: **Dr. Erand Llanaj, MTA-DE Public Health Research Group of the Hungarian Academy of Sciences, University of Derbecen

---

## [Editor Report · Acceptance letter]

4 Apr 2022

PONE-D-21-23425R1 

Resemblance of nutrient intakes in three generations of parent-offspring pairs: Tehran lipid and Glucose Study 

Dear Dr. Hosseini-Esfahani:

I'm pleased to inform you that your manuscript has been deemed suitable for publication in PLOS ONE. Congratulations! Your manuscript is now with our production department. 

Kind regards, 

on behalf of

Dr. Samson Gebremedhin 

Academic Editor

PLOS ONE